# Metformin Alleviates Left Ventricular Diastolic Dysfunction in a Rat Myocardial Ischemia Reperfusion Injury Model

**DOI:** 10.3390/ijms21041489

**Published:** 2020-02-21

**Authors:** Woori Jo, Kyung-Ku Kang, Sehyun Chae, Woo-Chan Son

**Affiliations:** 1Laboratory Animal Center, Daegu-Gyeongbuk Medical Innovation Foundation, Daegu 41061, Korea; c2dar@dgmif.re.kr (W.J.); kangkk@dgmif.re.kr (K.-K.K.); 2Department of Pathology, Asan Medical Institute of Convergence Science and Technology, Asan Medical Center, University of Ulsan College of Medicine, Seoul 05505, Korea; 3Korea Brain Bank, Korean Brain Research Institute, Daegu 41062, Korea

**Keywords:** acute myocardial infarction, coronary artery ligation, myocardial ischemia/reperfusion injury, systolic function, diastolic function

## Abstract

An increased incidence of myocardial infarction (MI) has recently emerged as the cause of cardiovascular morbidity and mortality worldwide. In this study, cardiac function was investigated in a rat myocardial ischemia/reperfusion (I/R) model using echocardiography. Metformin administration significantly increased ejection fraction and fractional shortening values on Days 3 and 7 when MI occurred, indicating that metformin improved left ventricular systolic function. In the Sham + MET and MI + MET groups, the E’ value was significantly different up to Day 3 but not at Day 7. This may mean that left ventricular diastolic function was effectively restored to some extent by Day 7 when metformin was administered. These results suggest that diastolic dysfunction, assessed by echocardiography, does not recover in the early phase of ischemic reperfusion injury in the rat myocardial I/R model. However, administering metformin resulted in recovery in the early phase of ischemic reperfusion injury in this model. Further gene expression profiling of left ventricle tissues revealed that the metformin-treated group had notably attenuated immune and inflammatory profiles. To sum up, a rat myocardial I/R injury model and ultrasound-based assessment of left ventricular systolic and diastolic function can be used in translational research and for the development of new heart failure-related drugs, in addition to evaluating the potential of metformin to improve left ventricular (LV) diastolic function.

## 1. Introduction

Myocardial infarction (MI) is the leading cause of morbidity and mortality worldwide. Acute myocardial infarction (AMI) is strongly related to metabolic syndrome, including dyslipidemia, smoking, diabetes mellitus, and hypertension [1], and it accounts for 30% of the total MI mortality. MI is characterized by an interruption of the blood supply to the left ventricle (LV), resulting in ischemic damage to the heart muscle [2,3]. Myocardial dysfunction in the ischemic state occurs prior to the appearance of chest pain or electrocardiogram (ECG) alteration [4]. Left ventricular function is the best indicator of prognosis in MI patients [5], and patients with left ventricular diastolic dysfunction have poorer surgical outcomes than patients with left ventricular systolic dysfunction in the perioperative period [6]. Therefore, echocardiography is a well-established, non-invasive diagnostic tool for the accurate evaluation of myocardial dysfunction, cardiac anatomy and hemodynamic function in clinical practice [7].

Rodents remain the most commonly used animal models in cardiovascular research [8]. Although rats have a rapid heart rate of >300 beats per minute, recent technological advancements mean that echocardiography can be used as a basic research tool for the evaluation of cardiac function in laboratory animals [8,9,10]. In addition, a rat myocardial ischemia/reperfusion (I/R) injury model has recently been developed, which has been widely used to evaluate new drugs or stem cell therapies [11,12,13,14].

Metformin, a dimethyl biguanide, is a first-line oral treatment for patients with type II diabetes, as directed by the guidelines of the American Diabetes Association (ADA) and the European Association for the Study of Diabetes (EASD) [15]. The cardio-protective action of metformin cannot be entirely attributed to its anti-hyperglycemic actions [16,17], and a short-term treatment study of metformin using an isoproterenol-induced myocardial infarction rat model has been conducted to assess the mechanism [18]. However, it is necessary to evaluate cardiac function in the early phase after MI because I/R injury is a common cause of AMI; furthermore, therapeutic strategies for the prevention of myocardial I/R injury can improve clinical outcomes in patients [2,3,12,14].

This study aimed to show the effectiveness of metformin on left ventricular diastolic function using echocardiography in a rat myocardial I/R injury model.

## 2. Results

### 2.1. Body Weight and Heart Changes

A representative gross photograph of a heart 7 days after MI surgery is shown in Figure 1B. The total body weight change over 7 days (%) is shown in Figure 1C,D; the body weight difference in groups B and D (induced MI) was lower than that in groups A and C on Day 7. In particular, there was a significant difference between groups B and C versus group A (*p* < 0.05 and *p* < 0.01, respectively). As seen in Figure 1E,F, assessment of heart weight changes at Day 6 and the heart weight/body weight (%) at Day 7 showed that the relative heart weight (%) was highest in group D (MI + MET), and significant differences were seen between group C (Sham + MET) and D (*p* < 0.01).

### 2.2. Echocardiographic Results

Echocardiographic results and the representative echocardiographic images are shown in Figure 2 and Table 1. The EF and FS values (reflecting LV systolic function) of the MI group were significantly decreased compared with those of the Sham group during the entire experimental period. Metformin administration improved the left ventricular systolic function, with rats in group D exhibiting significantly increased EF and FS values on Days 3 and 7 compared with group B. E’ values in the medial annulus (reflecting early active diastolic relaxation of the LV, an indicator of left ventricular diastolic dysfunction) were significantly decreased in group B compared with group A during the entire experimental period. This means that left ventricular diastolic dysfunction had not recovered at Day 7 after MI surgery. However, in group C (Sham + MET) and D (MI + MET), E’ values were significantly different up to Day 3 but not at Day 7. This may mean that metformin administration restored left ventricular diastolic dysfunction to some extent by Day 7. In addition, the E/E’ ratio reflects the mean left atrial pressure, indicating LV filling pressure, and the elevated values are an important indicator of poor prognosis in humans. In this experiment, the E/E’ ratio was significantly different up to Day 7 in groups A (Sham) and B (MI), meaning that left ventricular diastolic dysfunction did not recover during the entire experimental period. The differences between groups B (MI) and D (MI + MET) show the impact of metformin administration, and were significant throughout the experiment. In addition, comparison of groups C (Sham + MET) and D (MI + MET) showed no significant differences between the Sham and MI groups during the study period. This indicates that metformin administration could prevent left ventricular diastolic dysfunction caused by MI, particularly in the early phase.

### 2.3. Myocardial Infarct Size and Histopathological Analysis

Representative gross morphology of representative cardiac tissue sections stained with 1% TTC are shown in Figure 3A. Myocardial infarct size is decreased in group D (MI + MET: 24.45 ± 5.19) compared with group B (MI: 33.05 ± 6.15). However, there was no statistical significance. Representative photomicrographs of tissue sectioned with hematoxylin and eosin and Masson’s trichrome are also shown in Figure 3D–F. Continued coagulative necrosis, inflammatory cells infiltration and endocardial fibrosis is shown in the hematoxylin and eosin-stained slides. The Masson’s trichrome stain shows areas of marked fibroblast and collagen deposition. These histopathologic scoring is shown in Table 2 and Appendix A. In the apex area, metformin administration improved the advanced lesion with statistical significance. Figure 4B,C compares areas of collagen fibers in the LAD-ligation area of the LV (parasternal SAX area with papillary muscle) and the apex area. The degree of fibrosis in both the SAX and apex regions in group B (MI: 18.42 ± 3.59, *n* = 3; 23.72 ± 4.06, *n* = 3, in SAX and apex, respectively) was higher than that seen in group D (MI + MET: 10.80 ± 6.61, *n* = 3; 17.23 ± 12.49, *n* = 3, respectively), although these differences did not reach statistical significance.

### 2.4. Metformin Treatment Results in Changes of Multiple Cellular Processes in the Rat Myocardial I/R Injury Model

To investigate genes affected by metformin, gene expression profiling of the A (Sham), B (MI) and D (MI + MET) group was performed and a total of 2924 differentially expressed genes (DEGs) were identified (Materials and Methods) from the comparisons (Figure 4A and Appendix A): 2501 DEGs (1332 up-regulated and 1169 down-regulated) in MI samples, compared to the Sham controls (group B versus group A); and 1026 DEGs (493 up-regulated and 533 down-regulated) in the metformin-treated MI samples, compared to MI (group D versus group B). Of the 2924 DEGs, 603 (20.6%) were shared between two comparisons while the other DEGs were uniquely changed in the individual comparisons.

To identify reliable set of metformin associated genes, the DEGs were categorized into 8 clusters (C1-8) based on their differential expressions in the two comparisons (Appendix A and Appendix A). Among the clusters, C2 (359 genes) and C3 (236 genes) showed up- and down-regulation in their abundances by MI, respectively, but inhibition of the alterations by treatment of metformin (Figure 4B). C1 and C4 showed up- and down-regulation by MI, respectively, but inhibition of the alteration by treatment of metformin was not statistically significant (Figure 4B). Here, we aimed to identify potential therapeutic targets to metformin for myocardial I/R. Thus, we focused on C2 and C3.

To understand cellular processes represented by these two clusters (C2 and C3), the enrichment analysis of gene ontology biological processes (GOBPs) were performed using DAVID software. The analysis revealed that the genes in C2 were mainly involved in immune/inflammation responses and apoptosis (Figure 4C), while C3 were in glucose and fatty acid metabolism and cardiovascular system development (Figure 4D). In particular, the genes involved in immune/inflammation responses were strongly restored in their expression by metformin (*p* < 0.01) (Figure 4C,E). To examine collective associations of up- and down-regulated signaling pathways, a network model describing interaction among the differentially expressed genes was reconstructed (Figure 4F). The network has shown that metformin treatment inhibits increased inflammation and apoptosis by lowering the Tgf-β/BMP signaling pathway Metformin has also been shown to promote cell proliferation and anti-apoptosis by increasing the Jak/Stat and AMPK-PGC1α signaling pathways (Figure 4F,G) [19]. Taken together, these data suggest promising positive effects of metformin on myocardial I/R injury, consistent with previous findings [20,21].

## 3. Discussion

In this in vivo model, the positive effect of metformin on the early stage of MI was confirmed by assessing total weight change over the study period, the relative heart weight, left ventricular systolic and diastolic function using echocardiography on Days 1, 3, and 7 as well as the degree of fibrosis. These data demonstrate that metformin successfully attenuated left ventricular diastolic dysfunction induced by MI in this animal model.

MI, including AMI, remains the leading cause of morbidity and mortality worldwide [1,2,3], and left ventricular diastolic dysfunction during AMI has emerged as an important indicator of poor surgical outcomes and recurrence [6]. For human MI patients, left ventricular function is the best indicator of prognosis and is a well-established diagnostic tool [5,7]. In the current study, it was possible to use echocardiography for the accurate evaluation of cardiac function due to recent technological advancements [8,9,10]. Previous studies of the rat myocardial I/R injury model have involved relatively long-term exposure of metformin, for at least 2 weeks, and have rarely examined left ventricular diastolic function using echocardiography [11,12,13,14]. In the present study, the rat myocardial I/R injury model was successfully induced by transient ligation of the LAD, and metformin was administered 3 days prior to surgery to 7 days post-surgery, which is a relatively short exposure period compared with previous studies. With the use of echocardiography, we found that the rats in group B (MI) exhibited decreased EF and FS values, reflecting LV systolic dysfunction, and also showed reduced E’ values and increased E/E’ values, reflecting LV diastolic dysfunction, which resembles human MI. Since E’ and E/E’ values reflect the global function of the LV and are relatively independent of LV systolic function, heart rhythm abnormalities and LV hypertrophy [22,23], the E/E’ ratio is an acceptable reflection of mean left atrial pressure and, therefore, LV filling pressure [24]. Furthermore, in humans, elevation of LV filling pressure is the key indicator of poor outcomes, such as mortality, morbidity and length of stay in the ICU/hospital [25,26]. Therefore, evaluation of left ventricular diastolic function is important in MI patients, and confirming this is a good indicator in prognostic prediction and efficacy evaluation of new drug development effective for heart disease.

In humans, the pathologic consequences of MI are mainly seen in the coronary arteries and myocardium [27]. Early post-MI changes and MI lesions can be classified over time under light microscopy. Previous studies have demonstrated various types of lesions in humans using hematoxylin and eosin staining, and advanced lesions were also seen in the current rat myocardial I/R injury model. For example, continued coagulative necrosis was observed on Day 7 in the rat MI samples, which may occur in humans on Days 3–7. In addition, lesions resulting from macrophage phagocytosis, nuclear dust and endocardial fibrosis can occur in humans at 2–3 weeks, and can be identified as complex lesions in the same sample. Furthermore, Masson’s trichrome staining is widely used to study a range of pathologies in the muscle (muscular dystrophy), heart (infarct), liver (cirrhosis) or kidney (glomerular fibrosis), to confirm the presence of connective tissue. Here, we also identified the degree of fibrosis in the LV.

Metformin is a first-line anti-diabetic agent [15], and this study is intended to identify the effects of metformin on AMI which is highly associated with metabolic syndrome [1]. A previous mechanism study was conducted using isoproterenol-induced MI in a rat model [18], but the cardio-protective action of metformin is not attributed entirely to its anti-hyperglycemic actions [16,17]. The recent development of new formulations of metformin has improved its efficiency and tolerability, thereby expanding its clinical applications, i.e., allowing drug repositioning to exploit its non-glycemic effects [28,29,30,31,32,33]. A previous study outlined the non-glycemic effects of metformin, including the potential to improve cardiovascular clinical outcomes, anticancer effects and longevity [34]. In addition, the United Kingdom Prospective Diabetes Study (UKPDS) noted that intensive blood-glucose control with metformin lowered the risk of MI by 39% over a period of 10 years [35], and the promising results of the UKPDS clinical trial have supported the protective effects of cardiovascular complications of diabetes mellitus [36]. Other studies showed that patients receiving intensive glycemic management with sulfonylurea or insulin did not exhibit improved cardiovascular outcomes and therefore the potential cardiovascular effects of metformin treatment appear to be independent of glycemic control [29,37]. In studies related to longevity, metformin treatment has shown a possible effect on extending life span; indeed, the dietary restriction mechanism of metformin has long been known to increase life span [38,39,40,41]. The PRESTO Trial highlighted the relevance of metformin therapy, demonstrating that treatment resulted in decreased rates of death and MI in patients with diabetes undergoing percutaneous coronary intervention [42]. There are an increasing number of prospective clinical trials evaluating the use of metformin in the treatment of cardiovascular disease, and a reduction in all-cause mortality has been observed in patients with CKD and chronic HF [37], which may lead to expanded indications for metformin in disorders presently thought to be contraindications.

The current study highlighted the positive effects of metformin on acute MI; the weight loss associated with metformin treatment was confirmed, and the increased heart weight/body weight (%) in the MI-induced groups reflects cardiac hypertrophy, as reported in previous studies [28]. Echocardiographic data showed that left ventricular cardiac dysfunction does not recover in the early phase of ischemic reperfusion injury in this rat model throughout the experimental period, but metformin administration significantly improved left ventricular systolic function on Day 7. In addition, E/E’ values, an important indicator of cardiac diastolic function and poor outcomes in humans, significantly improved in the early stage of MI onset when metformin was administrated. There are some limitations to using E’ and E/E’, which show only the global function of the LV. However, further studies are planned using quantitative evaluation of regional function and the filling dynamics of the LV using strain, strain rate and speckle tracking with color tissue doppler [43,44].

The alteration of molecular signatures by metformin in rat myocardial I/R injury model has not been systematically explored despite its clinical importance. In this study, gene expression profiles of rat myocardial I/R injury model were examined. Our approach of transcriptome profiling provides a list of DEGs, thus extending the current list identified by conventional experiments. Furthermore, the gene ontology biological processes (GOBP) and network model can provide a molecular basis for understanding immune/inflammation and cardiovascular system alteration in myocardial I/R. The network model further suggested fatty acid metabolism, mitochondrial biogenesis, Tgf-β/BMP and Jak/Stat signaling pathways associated with positive effects of metformin in a myocardial I/R injury. To the best of our knowledge, the current study is the first to demonstrate the promising positive effects of metformin on left ventricular diastolic dysfunction in a rat myocardial I/R injury model using echocardiography. The rat myocardial I/R injury model is a good representation of human AMI, allowing the cardio-protective effects of metformin to be evaluated. These data provide additional understanding of the effects of metformin on the development of AMI and provide a clear rationale for the use of metformin in high-risk patients.

## 4. Materials and Methods

### 4.1. Animals, Husbandry and Experimental Design

Sprague–Dawley rats (8-week-old adult males, mean weight: 287.90 ± 6.62 g) were purchased from Koatech (Kyungki, South Korea); the study design was approved by the Institutional Animal Care and Use Committee of Daegu-Gyeongbuk Medical Innovation Foundation (DGMIF-19022001-00, 17 02 2019). The rats were housed at a temperature of 22 ± 1°C and relative humidity of 50% ± 10%, using 12 h light/dark cycles, illumination at 150–300 Lux and with ventilation 10–20 times/hour. These conditions were monitored every hour for 24 h and maintained within an acceptable range throughout the study. The rats were housed three per cage at the beginning of the study and fed an autoclaved pellet diet (SAFE + 40RMM; SAFE Diets, Augy, France) ad libitum.

The experimental design is shown in Figure 1A. The rats were divided into four groups (six rats/group) as follows: A) Sham group, B) MI group, C) Sham + MET group and D) MI + MET group. The rats in groups A and B received reverse osmosis (RO) drinking water, and the rats in groups C and D were treated with 200 mg/kg metformin (Sigma-Aldrich, St Louis, MO, USA) dissolved in RO drinking water by oral injection daily for 11 days (3 days before the surgery to 7 days post-surgery).

### 4.2. Induction of Myocardial Ischemia/Reperfusion Injury

Animals were anesthetized with alfaxlaone (50 mg/kg, IP) and xylazine (5 mg/kg, IP). After anesthetization, rats were intubated and ventilated by respirator (Harvard Apparatus Inspira, MA, USA) with a tidal volume of 3.0 mL/kg and at a rate of 60 bpm. Throughout the surgery, the rats were placed on a heated plate and monitored by ECG [24]. The rat myocardial I/R injury model involved ligation of the left anterior descending (LAD) coronary artery for 30 min, as described previously [24]. MI was confirmed by the paleness of the apical region of the heart and elevation of the S–T segment on the ECG [24]. The Sham group underwent the same surgical procedure without ligation.

### 4.3. Echocardiographic Analysis

Echocardiography was performed on Days 1, 3 and 7 after the induction of MI, using Vevo2100 (VisualSonics Inc., Ontario, Canada). The rats were anesthetized with alfaxalone (50 mg/kg, IP) and xylazine (5 mg/kg, IP), and were placed in a supine position. They were monitored by ECG and maintained at 37 °C.

Echocardiographic parameters were recorded in accordance with the American Society of Echocardiography guidelines [1]. Images were obtained from the left parasternal short-axis (SAX) view of the LV at the level of papillary muscles to define wall thicknesses and internal diameters during systole and diastole, and regional wall motion abnormalities in the LAD territory. The left ventricular posterior wall at the diastolic and systolic phase, and the interventricular septum thicknesses at the diastolic and systolic phase, were measured. In addition, the left ventricular internal diameter at the diastolic and systolic phase, ejection fraction (EF), fractional shortening (FS), stroke volume and cardiac output were measured for the systolic function of the LV. At the apical four chamber view, the E/A and E/E’ ratios were calculated using color flow doppler, pulsed wave doppler and tissue doppler imaging for the diastolic function.

### 4.4. Myocardial Infarct Size and Histopathological Analysis

After echocardiographic evaluation on Day 7, the rats were euthanized under isoflurane anesthesia and their hearts were excised. Following reperfusion with 0.9% normal saline, the hearts were sectioned into 2 mm transverse slices and the slices were immersed in 1 % solution of 2,3,5-triphenyltetrazolium chloride (TTC, Sigma, St.Loius, MO) at 37 °C for 15 min in the dark. The images of these slices were obtained using a digital camera. The infarct area and total area were analyzed by the Image J program (provided by NIH). For the histopathological analysis, the heart tissue was fixed in 10% neutral buffered formalin (BBC Biochemicals, Mount Vernon, WA, USA). A tissue processor (Thermo Fisher Scientific, Inc., Runcorn, UK) was used to prepare samples from the formalin-fixed tissue for analysis by fixing, dehydrating and staining. The paraffin-embedded tissue blocks were cut at a 4 μm thickness and mounted onto glass slides. Staining was performed with hematoxylin (YD-Diagnostics, Kyungki, Korea) and eosin (BBC Biochemicals, Mount Vernon, WA, USA) using an autostainer (Dako Coverstainner; Agilent, Santa Clara, CA, USA). In addition, tissue sections were stained using a Masson’s trichrome staining kit according to the manufacturer’s instructions (ScyTek Laboratories, West Logan, UT, USA). After staining, parasternal short axis with papillary muscle and apex area of the left ventricle were scanned with a slide scanner (Pannoramic SCAN II; 3DHISTECH, Budapest, Hungary) and were captured by slide viewer (CaseViewer; 3DHISTECH). The morphometric analysis of fibrosis was performed using image J program (provided by the NIH). The red and blue color was designated in the MT stained section, and the blue area (collagen fiber) was measured compared to the total red area. Histopathological inspection was performed in a blinded manner by two investigators.

### 4.5. mRNA Sequencing

For RNA sequencing, total RNAs were obtained from left ventricle tissues of three rat groups (Sham group, MI group, and MI + MET group) using Trizol reagent (Invitrogen Life Technologies, Grand Island, NY). The integrity of the total RNA was analyzed using an Agilent Bioanalyzer. The RNA integrity values for all of the samples were larger than 7. Poly (A) mRNA isolation from total RNA and fragmentation were performed using the Illumina TruSeq Stranded mRNA Sample Prep Kit, according to the manufacturer’s instructions. The adaptor ligated libraries were sequenced using an Illumina NovaSeq 6000 (Bioneer, Korea). In each condition, the mRNA-sequencing analysis was performed for two biological replicates obtained from independent rats. From the resulting read sequences for each sample were aligned to the Rattus_norvegicus reference genome (Rnor_6.0) using STAR software (version 2.7) with the default parameters [45]. After the alignment, the numbers of reads mapped to the gene features (GTF file of Rnor_6.0.90) were counted using HTseq [46]. The read counts for the samples in each condition were then normalized using the TMM (trimmed mean of M-values) normalization of the edgeR package [47] and converted to log_2_-values.

### 4.6. Analysis of mRNA Sequencing Data

To identify DEGs between three conditions, the previously reported statistical hypothesis test was performed [48]. Briefly, for each gene, a T-statistic value was calculated using Student’s t-test in each of the two comparisons (MI group versus Sham group or MI + MET group versus MI group). For each comparison, the empirical distributions of T-statistic value for the null hypothesis (i.e., the genes are not differentially expressed) were estimated by performing all possible combinations of random permutations of samples. Using the estimated empirical distributions, adjusted P-values for Student’s t-test for each gene were computed. Finally, the DEGs were identified as the ones that have the adjusted *p*-values ≤ 0.05 and absolute log2-fold-changes ≥ 0.58 (1.5-fold). To identify cellular processes represented by the DEGs, the enrichment analysis of GOBPs was performed using DAVID software [49] and the GOBPs with a P-value < 0.05 were selected as the processes enriched by the DEGs.

### 4.7. Statistical Analysis

Statistical significance was determined using GraphPad Prism 6 (GraphPad Software, San Diego, CA, USA). All data are presented as mean ± standard deviation (SD). The results for each group were compared by one-way analysis of variance (ANOVA) with Tukey’s multiple comparisons test, Dunnett’s multiple comparisons test and a two-tailed unpaired t-test. A *P* < 0.05 was considered statistically significant.

## Figures and Tables

**Figure 1 ijms-21-01489-f001:**
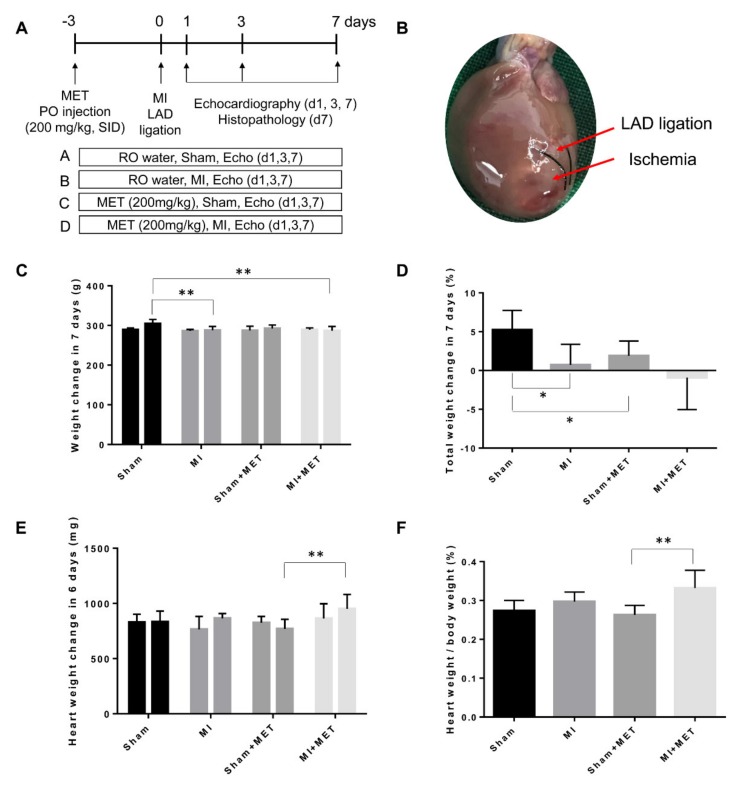
Experimental design using a rat myocardial infarction/reperfusion injury model. (**A**) The animals were divided into four groups: Group A, Sham; Group B, MI; Group C, Sham + MET; Group D, MI + MET. (**B**) Representative gross photograph of a heart 7 days after MI surgery. (**C**) Body weight change in 7 days. (**D**) Differences in total body weight change in 7 days. (**E**) Heart weight change in 6 days. (**F**) Heart weight/body weight (%) on Day 7. * indicates a statistically significant difference compared with the Sham group by two-sample *t*-test (*p* < 0.05). ** indicates a statistically significant difference compared with the Sham group (*p* < 0.01) by two-tailed unpaired t-test (**D**,**F**) and Tukey’s multiple comparisons test (**C**,**E**). LAD, left anterior descending; MET, metformin; MI, myocardial infarction; PO, per oral; RO, reverse osmosis.

**Figure 2 ijms-21-01489-f002:**
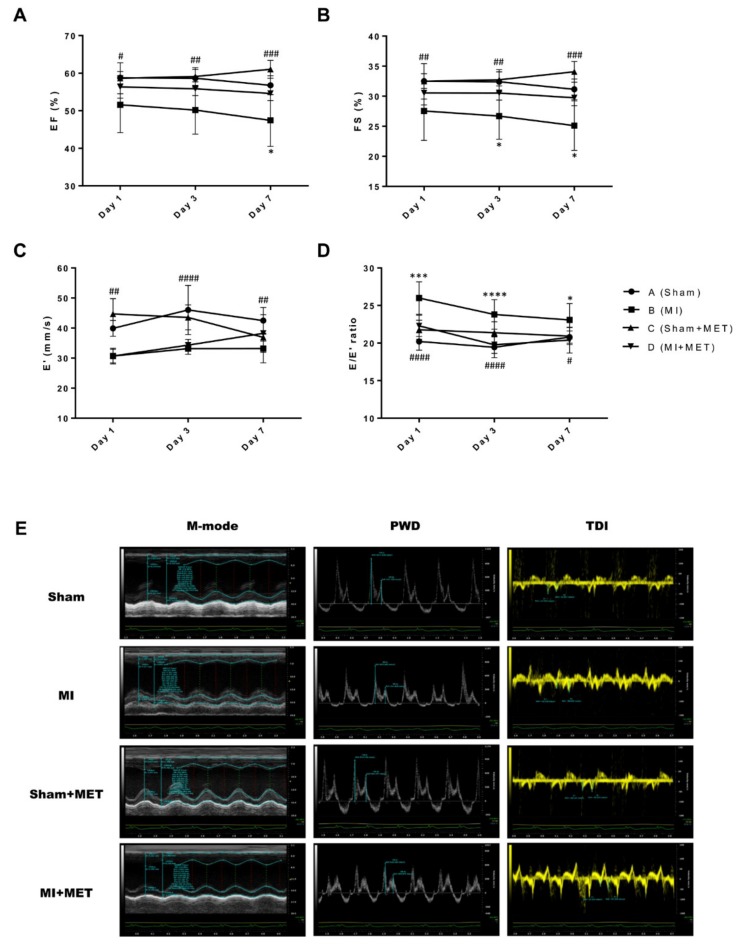
Echocardiographic results and representative echocardiographic imaging of rat hearts on Day 3 after MI surgery. Metformin administration significantly improved LV diastolic function. (**A**) Ejection fraction, EF; (**B**) Fractional shortening, FS; (**C**) Early relaxation velocity on tissue doppler, E’; (**D**) E/E’ ratio of diastolic function. EF and FS determined by M-mode imaging reflects LV systolic dysfunction; the E’ and E/E’ ratio determined by pulse wave doppler and tissue doppler imaging reflects LV diastolic dysfunction. (**E**) representative echocardiographic imaging. * *p* < 0.05, *** *p* < 0.001, **** *p* < 0.0001 between group B and D; # *p* < 0.05, ## *p* < 0.01, ### *p* < 0.001, #### *p* < 0.0001 between group A and D by Dunnett’s multiple comparisons test compared with MI group. PWD, Pulse Wave Doppler; TDI, Tissue Doppler Imaging.

**Figure 3 ijms-21-01489-f003:**
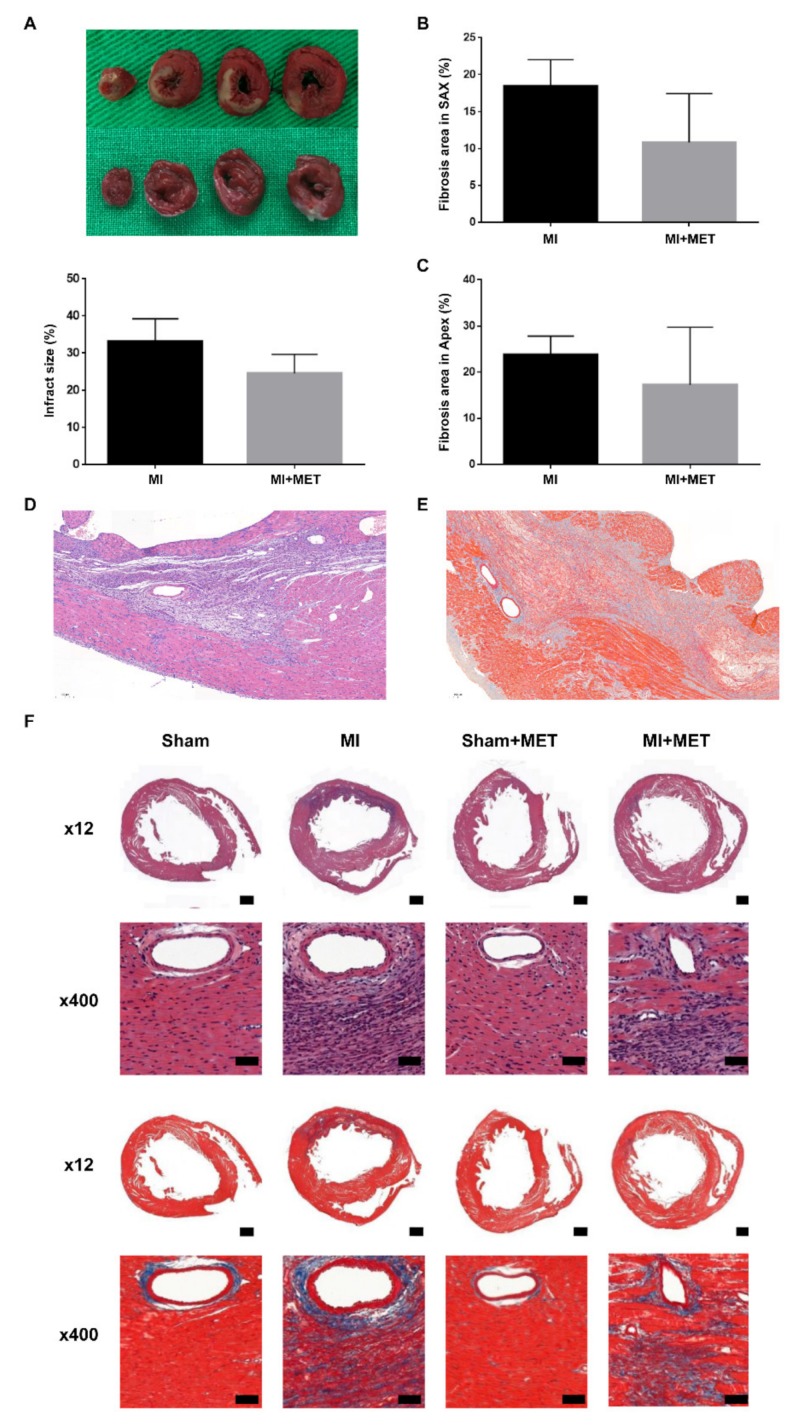
Protective effect of metformin on infarct size and histopathological changes in the left ventricle of a rat. (**A**) Representative photogram of left ventricle slice and the infarct size; (**B**–**C**) degree of fibrosis (%) in the SAX (short-axis region with papillary muscle) and apex area (*n* = 3 in both groups); and (**D**–**F**) hematoxylin and eosin and Masson’s trichrome staining on Day 7 (**D**,**E**,: ×100; F, scale bar: 1000 μm, 50 μm).

**Figure 4 ijms-21-01489-f004:**
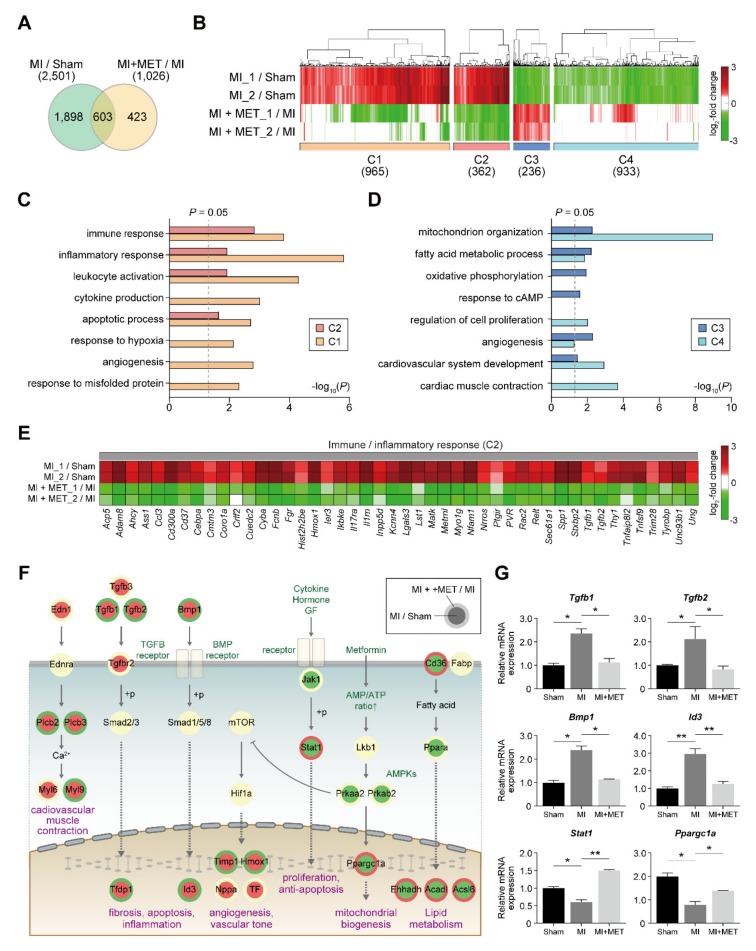
Cellular processes affected by metformin treatment. (**A**) Relationships among DEGs from the two comparisons (Group B/Group A and Group D/Group B). (**B**) Clusters (C1-4) of genes affected by metformin. Red and green denote up- and down-regulation, respectively. Color bar means gradient of log2-fold-changes. Numbers of DEGs in the comparisons of clusters are denoted in parenthesis (**A**–**B**). (**C**–**D**) Cellular processes enriched by DEGs in C1-4. X-axis, −log10(P) where *p* is the enrichment *p*-value calculated in DAVID software. (**E**) DEGs in C2 involved in the immune/inflammatory response. (**F**) Network model describing interactions among signaling pathways. Arrows and suppression symbols, activation and inhibition in signaling. “+p”, phosphorylation. Red and green denote up- and down-regulation, respectively. (**G**) Gene expression levels obtained from mRNA-seq. The data are presented as mean ± SD. For each gene, all the significant differences (see Materials and Methods) from the two comparisons were indicated by an asterisk: * *p* < 0.05 and ** *p* < 0.01.

**Table 1 ijms-21-01489-t001:** Cardiac function measured by echocardiography.

Cardiac Function	Day 1				Day 3				Day 7			
	Sham	MI	Sham + MET	MI + MET	Sham	MI	Sham + MET	MI + MET	Sham	MI	Sham + MET	MI + MET
EF, %	*58.79 ± 1.65	51.60 ± 7.43	*58.64 ± 4.14	56.36 ± 3.01	**58.65 ± 2.38	50.17 ± 6.37	**59.12 ± 2.37	55.86 ± 1.82	***56.79 ± 2.52	47.44 ± 6.92	****61.04 ± 2.39	*54.63 ± 1.91
FS, %	**32.52 ± 1.22	27.55 ± 4.90	**32.48 ± 2.94	30.56 ± 2.00	**32.41 ± 1.65	26.70 ± 3.86	***32.73 ± 1.70	*30.54 ± 1.18	***31.16 ± 1.71	25.11 ± 4.13	****34.70 ± 1.72	*29.73 ± 1.31
SV, µL	****237.16 ± 25.70	162.13 ± 20.91	162.97 ± 13.01	163.64 ± 14.13	**228.59 ± 19.43	185.49 ± 22.62	**227.65 ± 2037	*218.79 ± 18.72	235.95 ± 24.19	208.30 ± 21.03	222.30 ± 26.28	**247.02 ± 29.44
CO, mL/min	****62.42 ± 7.03	50.42 ± 13.71	63.80 ± 9.31	55.77 ± 7.38	**60.82 ± 7.43	48.30 ± 8.02	**63.81 ± 5.59	*69.43 ± 10.22	60.58 ± 6.73	63.23 ± 23.69	56.10 ± 9.15	**63.17 ± 8.30
LVIDd, mm	*8.38 ± 0.31	**7.60 ± 0.53	8.50 ± 0.35	7.36 ± 0.33	8.33 ± 0.41	8.30 ± 0.56	8.36 ± 0.39	8.38 ± 0.32	8.58 ± 0.54	8.81 ± 0.85	*8.08 ± 0.52	8.92 ± 0.39
LVIDs, mm	5.70 ± 0.31	5.46 ± 0.73	5.89 ± 0.51	5.11 ± 0.49	5.60 ± 0.36	6.00 ± 0.84	5.73 ± 0.28	5.82 ± 0.29	*5.92 ± 0.58	6.74 ± 0.91	***5.53 ± 0.43	6.35 ± 0.34
IVSd, mm	1.44 ± 0.17	1.51 ± 0.16	1.41 ± 0.16	1.68 ± 0.24	1.39 ± 0.13	1.29 ± 0.05	1.37 ± 0.15	1.47 ± 0.24	1.40 ± 0.21	1.49 ± 0.50	1.38 ± 0.12	1.45 ± 0.15
IVSs, mm	2.30 ± 0.24	1.98 ± 0.22	2.18 ± 0.15	*2.48 ± 0.30	2.22 ± 0.24	1.94 ± 0.28	2.32 ± 0.24	*2.44 ± 0.43	2.33 ± 0.32	2.10 ± 0.37	2.21 ± 0.22	2.26 ± 0.44
LVPWd, mm	1.58 ± 0.17	1.60 ± 0.17	1.58 ± 0.15	**1.83 ± 0.09	1.50 ± 0.09	1.56 ± 0.04	1.53 ± 0.11	1.66 ± 0.14	1.53 ± 0.15	1.63 ± 0.14	1.57 ± 0.09	1.64 ± 0.09
LVPWs, mm	2.36 ± 0.19	2.27 ± 0.19	2.33 ± 0.19	*2.56 ± 0.18	*2.50 ± 0.19	2.23 ± 0.17	2.29 ± 0.18	*2.49 ± 0.16	2.27 ± 0.25	2.11 ± 0.18	*2.39 ± 0.20	*2.41 ± 0.12
E’, mm/s	**39.90 ± 2.63	30.67 ± 2.24	****44.70 ± 5.09	30.67 ± 2.59	****46.00 ± 8.18	33.14 ± 1.89	***43.53 ± 4.14	34.31 ± 1.91	**42.49 ± 4.35	33.20 ± 4.78	36.91 ± 1.27	38.21 ± 6.22
E/A ratio	1.69 ± 0.25	1.92 ± 0.24	2.08 ± 0.80	1.46 ± 0.26	2.16 ± 0.66	2.34 ± 0.41	1.69 ± 0.29	1.56 ± 0.30	2.58 ± 0.90	1.81 ± 0.42	1.54 ± 0.17	2.05 ± 1.49
E/E’ ratio	****20.20 ± 1.15	25.99 ± 2.15	****21.77 ± 1.27	***22.29 ± 1.42	****19.44 ± 1.39	23.81 ± 1.97	*21.37 ± 1.46	****19.77 ± 1.15	*20.79 ± 0.82	23.08 ± 2.18	*20.93 ± 1.14	*20.39 ± 1.72

EF, ejection fraction; FS, fractional shortening; SV, stroke volume; CO, cardiac output; LVIDd, left ventricular internal diameter at diastole; LVIDs, left ventricular internal diameter at systole; IVSd, interventricular septal thickness at diastole; IVSs, interventricular septal thickness at systole; LVPWd, left ventricular posterior wall thickness at diastole; LVPWs, left ventricular posterior wall thickness at systole; E’, early diastolic tissue doppler velocity; E/A, the ratio of the early (E) to late (A) ventricular filling velocities; E/E’, the ratio of the early (E) to early diastolic tissue Doppler velocities. *(*p* < 0.05), ** (*p* < 0.01), *** (*p* < 0.001), **** (*p* < 0.0001) indicates a statistically significant difference by multiple comparison with Dunnett’s correction compared with MI group.

**Table 2 ijms-21-01489-t002:** Incidence of histopathological changes in the hearts of rats.

Histopathological Change	MI		MI + MET	
	SAX	Apex	SAX	Apex
Lesions (*n*)	3	3	3	3
Coagulation necrosis, myocardial	3.33 ± 0.58	3.33 ± 0.58	2.67 ± 0.58	2.67 ± 0.58
Inflammatory cells infiltration, epicardial	2.33 ± 0.58	2.67 ± 1.15	2.00 ± 1.00	1.67 ± 0.58
Inflammatory cells infiltration, myocardial	3.33 ± 0.58	3.67 ± 0.58	2.67 ± 0.58	3.33 ± 0.58
Inflammatory cells infiltration, endocardial	2.33 ± 0.58	1.67 ± 0.58	1.33 ± 0.58	1.33 ± 0.58
Fibrosis, epicardial	2.67 ± 1.15	2.33 ± 1.53	2.33 ± 1.15	2.00 ± 1.00
Fibrosis, myocardial	3.67 ± 0.58	3.67 ± 0.58	2.67 ± 0.58	2.33 ± 0.58
Fibrosis, endocardial	2.33 ± 0.58	2.00 ± 1.73	1.67 ± 1.15	1.00 ± 0.00
Total	2.86 ± 0.57	2.76 ± 0.81	* 2.19 ± 0.54	2.05 ± 0.80

Grading of histopathological changes in the rat left ventricle SAX with papillary muscle and apex area tissue. Grades 1, 2, 3 and 4 show minimal, slight, moderate, and severe pathological changes, respectively. Values are mean ± standard deviation (*n* = 3). * *p* < 0.05 indicates a statistically significant difference by two-tailed unpaired t-test compared with the MI group.

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
