# Peer review of "Metformin Alleviates Left Ventricular Diastolic Dysfunction in a Rat Myocardial Ischemia Reperfusion Injury Model"

_ijms, 2020, doi:10.3390/ijms21041489_

Round 1
Reviewer 1 Report
The enthusiasm was very low regarding this manuscript due to the lack of novelty as well as lack of mechanistic study.
There are several studies showed the effect of metformin on I/R with mechanistic studies:
https://www.ncbi.nlm.nih.gov/pmc/articles/PMC6528516/
https://www.ncbi.nlm.nih.gov/pmc/articles/PMC3132893/
https://diabetes.diabetesjournals.org/content/57/3/696.short
Author Response
[Response] We appreciate the reviewer for the helpful comments. In this study, we first demonstrate the positive effects of metformin on left ventricular diastolic dysfunction in a rat myocardial I/R injury model using echocardiography. However, as reviewer’s comments, our findings lack molecular evidences.
To address the Reviewer’s concerns, we performed mRNA sequencing in the left ventricle of the rats for the mechanism study, Sehyun Chae, a senior researcher at Korea Brain Research Institute, asked to analyze this data. We have performed gene expression profiling of left ventricle tissues from Sham, MI, and metformin treated MI samples. Through the analysis of mRNA-seq data, we identify differentially expressed genes and cellular changes in the following comparisons: MI versus Sham and MI+MET versus MI. We identified 2,924 genes whose expression levels were altered in MI. Among the 2,924 genes, 603 (20.6%) showed inhibition of alteration by metformin. Also, these genes were included in immune/inflammation response, apoptotic process, fatty acid metabolism associated with pathogenesis of MI. These results suggest the positive effects of metformin on MI.
Also, we would like to add a corresponding author and acknowledgment as a result of this contribution to the systems approach for mRNA sequencing. And we all authors agreed on this. Thank you for your consideration.
We added new data to Figure 4, Supplementary Figure S1, and table S1 and then revised the corresponding Results, Discussions, and Materials and Methods section shown as below.

Reviewer 2 Report
Great research! Congrats
Author Response
[Response] We appreciate the reviewer for the comments!
In this study, the main topic is the effectiveness of the metformin for early phase of left ventricular diastolic dysfunction in a rat myocardial ischemia reperfusion injury model.
In recent years, the importance of left ventricular diastolic dysfunction is emerging in human with poor prognosis than systolic dysfunction, and we think our results reflect this well. Thank you.

Reviewer 3 Report
In the current study, Metformin alleviates left ventricular diastolic dysfunction in a rat myocardial ischemia-reperfusion injury model the results were interesting, and with some alterations, the manuscript could be improved.
Which is the total n-number of rats were used? Could there be a sex effect and their structural changes? In the method section, the morphometric analysis of fibrosis must be described. The area, type, and size of myocardial infarction in this experimental model must be described, as well as the histological myocardial changes with MET treatment. The histological images do not clearly show the infarct zone and changes of cardiomyocytes in IM and MI+MET. The histological description of coagulative necrosis of cardiomyocytes and macrophage phagocytosis must be clearly visible. If authors indicate endocardial fibrosis they should demonstrate this part of the heart. Type of myocardial fibrosis should be described. The cardiac function measured by echocardiography is not a convincing indicator for this study, the cardiac markers must be included.Author Response
[Response] Thank you for your detailed review of the histopathology.
The number of animals used in the experiment was 6 per group as described in 4.1. Animals, husbandry and experimental design section. Echocardiography was examined in all rats. Three of the six per group were H & E and MT (Masson’s trichrome) stained for histopathological analysis as described, and the rest three per group were 1% TTC stained.
This experiment was conducted only in male rats, so we could not see the gender difference. However, given the increase in arteriosclerosis in postmenopausal women, further study about differences in gender may be valuable.
We also added the method section of the morphometric analysis of fibrosis and modified Figure3 to show the images clearly including infarct zone and histological changes. Area of myocardial infarction was shown in Figure 3A. Fibrosis was shown in Figure 3B and 3C. Histopathological evaluation was performed in more detail. The results were added to Figure 3D-3E and Table 2.
In line with your comments, we analyzed serum CK-MB and LDH, but unfortunately, the time point at 7 days after surgery was too late to show the result.
We revised Figure 3, Table 2, Supplementary table S1 Results and Materials and Methods section shown as below.

Round 2
Reviewer 1 Report
Thanks a lot for the authors in doing these changes to the manuscript which significantly improved the manuscript.
I have no further comments
Reviewer 3 Report
The manuscript has been improved and ready to publish.